# A Reliable and Standardizable Differential PCR and qPCR Methodology Assesses HER2 Gene Amplification in Gastric Cancer

**DOI:** 10.3390/biology10060516

**Published:** 2021-06-10

**Authors:** Ignacio Juarez, Juan Francisco Toro-Fernandez, Christian Vaquero-Yuste, Marta Molina-Alejandre, Inmaculada Lasa, Remedios Gomez, Adela Lopez, Jose Manuel Martin-Villa, Alberto Gutierrez

**Affiliations:** 1Department of Immunology, Ophthalmology and ENT, Facultad de Medicina, Universidad Complutense de Madrid (UCM), 28040 Madrid, Spain; ignajuar@ucm.es (I.J.); juanftor@ucm.es (J.F.T.-F.); cvaque01@ucm.es (C.V.-Y.); mamoli09@ucm.es (M.M.-A.); 2Instituto de Investigación Sanitaria Gregorio Marañón (IiSGM), 28007 Madrid, Spain; 3Hospital Universitario Príncipe de Asturias, 28006 Madrid, Spain; inmaculada.lasa@salud.madrid.org (I.L.); remedios.gomez@salud.madrid.org (R.G.); adelapetra.lopez@salud.madrid.org (A.L.); agutierrezcalvo@telefonica.net (A.G.)

**Keywords:** HER2, gastric cancer, cancer, PCR, immunotherapy

## Abstract

**Simple Summary:**

Patients with gastric cancer may present variations in the copy number of the HER2 gene in their primary tumors. The techniques used to detect these variations and HER2 overexpression render false positive and negative results with high frequency, and robust methodologies are required to assess this amplification and confidently select patients who may benefit from HER2-specific monoclonal antibody-based therapies. We addressed this issue by molecular biology techniques using DNA samples from tumor or distal tissue of gastric cancer patients. The HER2 and a control (IFNG) gene were subjected to differential (diffPCR) and quantitative PCR (qPCR). A cut-off point above which patients can be deemed positive was set based on the HER2/IFNG ratio, achieved using DNA from 30 healthy donors. Both, diffPCR and qPCR, identified the presence of somatic HER2 amplifications in 25% of patients in DNA from tumoral tissue, but not distal, paired tissue samples. Immunohistochemistry and immunofluorescence detected HER2 overexpression in tumor, but not distal, tissue of the patients previously identified as HER2+ by diffPCR and qPCR. Thus, the molecular biology-based techniques herein reported can identify patients with HER2 gene amplification and suitable for immune-based therapies.

**Abstract:**

We have applied two PCR techniques, differential PCR (diffPCR) and qPCR for the identification of HER2 gene amplifications in genomic DNA of tumor and distal gastric samples from patients with gastric cancer. The diffPCR technique consists of the simultaneous amplification of the HER2 gene and a housekeeping gene by conventional PCR and the densitometric analysis of the bands obtained. We established a cut-off point based on the mean and standard deviation analyzing the DNA of 30 gastric tissues from patients undergoing non-cancer gastrectomy. diffPCR and qPCR yielded consistent results. HER2-overexpression was detected in 25% of patients and was further confirmed by immunohistochemistry and immunofluorescence. The approaches herein described may serve as complementary and reliable methods to assess HER2 amplification.

## 1. Introduction

The HER2 (human epithelial growth factor receptor 2) protein is encoded for by the HER2/neu gene, located in the chromosomal region 17q12. This gene may suffer an increase in the gene copy-number, leading to HER2 overexpression [1]. As a result, uncontrolled cell growth and tumorigenesis take place. The use of monoclonal antibodies specific to HER2 (Trastuzumab) is relevant in the treatment of patients overexpressing this protein, and it then seems indispensable to have a robust and reliable procedure to identify patients able to benefit from this therapy. Several methodologies have been published, namely immunohistochemistry and FISH, requiring costly equipment and experienced readers. Moreover, both techniques are subjective and expensive [2,3]. In gastric cancer (GC), HER2 gene amplification would result in tumors having both greater invasive and proliferative capacity [4], and several authors have suggested that HER2 may involve poor outcome in patients with gastric cancer [5,6,7]. Although an increase in the HER2 gene copy-number has been demonstrated in 10% to 30% GC patients, different authors and meta-analysis differ in terms of the actual number of patients overexpressing the protein [7,8,9,10]. Most of these discordances can be explained by the subjectivity of the techniques used to assess HER2 gene amplification and overexpression (ISH and IHC), scoring criteria [11], tumor heterogeneity and tissue sampling errors [12]. Even with the discrepancies in the frequency of occurrence of HER2 amplifications in patients with gastric cancer, these mutations have, in addition to their previously mentioned prognostic value, a high predictive value, as it has been shown that therapy with anti-HER2 monoclonal antibodies generates good response rates in this type of patients [13,14].

Previous studies demonstrated that quantitative (qPCR) can be a useful approach to determine HER2 amplification in some types of cancer [15,16,17,18], but none focused on establishing a standardized and reproducible procedure enabling laboratories worldwide to assess a sample as negative or positive. In this work, we suggest the use of differential PCR (diffPCR) and qPCR as two complementary and easily standardizable techniques to analyze HER2 amplifications.

We wish to present two reliable and standardizable PCR-based methodologies to assess changes in the number of HER2 gene copies in tissue samples from GC samples, and compare the results obtained with HER2 protein overexpression evaluated by immunohistochemistry (IHC) and immunofluorescence (IF).

## 2. Materials and Methods

Fifty-six paired tissue samples (tumoral and distal) from twenty-eight gastric adenocarcinoma patients were used. Patients were classified in terms of age, sex, tumor stage, therapy, and progression (Table 1). Upon surgery, gastric tissue samples were obtained, snap-frozen in liquid nitrogen and stored at −80 °C until use. In addition, gastric tissue samples from individuals undergoing gastric surgery for reasons other than gastric tumor (i.e., morbid obesity) were obtained and processed likewise (n = 30). DNA from tissues was obtained with the Illustra Nucleon BACC Genomic DNA Extraction Kit, following manufacturer’s instructions. (GE Healthcare, Chicago, IL, USA). DNA quality and concentration were measured with a NanoDrop One (Thermo Scientific, Waltham, MA, USA).

HER2 copy number variations were assessed by two molecular biology approaches:

Differential PCR (diffPCR): HER2 and IFN-γ genes were co-amplified by PCR and then quantified by band densitometry in a 3% agarose gel. Two different pairs of HER2 primers, denoted “HER2 Primer A” and “HER2 Primer B”, were used to amplify the HER2 gene to have a double confirmation. The same set of primers for IFNG amplification were used in both PCRs. The primer sequences and amplicon sizes were as follows: The target gene (HER2) and a reference housekeeping gene with non-reported copy number variations 14 (IFNG) gene were co-amplified by PCR. Primer sequences, amplicon sizes and PCR conditions are shown in Table 2 [19].

As positive control, DNA from the cell lines SKBR-3 (metastatic breast adenocarcinoma) and BT-474 (invasive ductal carcinoma), known to present HER2 amplifications, was used. As negative control, the Caco-2 (colorectal adenocarcinoma) cell line was used.

Amplified samples were resolved by electrophoresis in a 3% agarose gel and the bands obtained subjected to densitometry (FIJI software). Bands’ intensities were normalized relative to the Caco-2 cell line band and a HER2/IFNG ratio for each individual was calculated (Figure 1).

Quantitative PCR (qPCR): Samples were amplified in a LightCycler^®^ 96 thermocycler using the same primers employed in diffPCR. The reaction mix was the commercial iQTM SYBR^®^ Green Supermix (Bio-Rad, Hercules, CA, USA), using a standard qPCR protocol. Quantification cycle (Cq) values were obtained for the paired patients’ samples, healthy controls, and cell lines. Relative quantification in real-time PCR was determined using the following Equation (1) by Pfaffl [20].
R = (E_HER2_) ∆Cq HER2 (control-samples)/(E_IFNG_) ∆Cq HER2 (control-sample)(1)

Being “E” the qPCR efficiency value for each of the amplified genes, “∆Cq” the difference in the number of thermal cycler to the threshold line, and “control” the Caco-2 line.

Cut-off establishment: To establish whether a sample could be deemed as positive, a HER2/IFNG ratio threshold was obtained as previously described in an in-house ELISA assay [21]. The threshold was established from the Cq and standard deviation (SD) values achieved with the twenty-five control samples as MeanCq + 2 * SD for diffPCR (CI 95%) and MeanCq + 3 * SD for qPCR (CI 99%). Individuals with a HER2/IFNG ratio value above the cut-off point in both methodologies (diffPCR and qPCR) were considered as positive.

Immunohistochemistry: HER2 expression was measured by Immunohistochemistry (IHC). We obtained 5 µm sections from the mucosa, air dried them and fixed them with acetone for 10 min, and then subjected them to immunohistochemistry staining. Endogenous HRP were blocked using Bloxall reagent for 10 min, and unspecific protein binding was blocked with 5% normal donkey serum. Mouse monoclonal antibody to HER2 (Abcam, 3B5 clone) at a concentration of 5 μg/mL was applied to the tissue section and left at 4 °C overnight. After a washing step with PBS, 2 μg/mL goat antimouse secondary antibody (Santa Cruz Biotechnology) was then used for one hour at room temperature. Staining was revealed with a DAB solution (1 mg/mL) and rinsed after 3 min. Samples were further counterstained with hematoxylin and mounted.

Immunofluorescence: HER2 expression also measured by Immunofluorescence (IF). We obtained 5 µm sections from the mucosa, air-dried them, and fixed them with acetone for 10 min, and then subjected to immunofluorescence staining and unspecific protein binding was blocked with 5% normal donkey serum. Mouse monoclonal antibody to HER2 (Abcam, 3B5 clone) at a concentration of 5 μg/mL was applied to the tissue section and left at 4 °C overnight. After a washing step with PBS, 2 μg/mL Alexa Fluor 594 AffiniPure Donkey Anti-Mouse IgG (Jackson ImmunoResearch, Cambridgeshire, UK) antibody was then used for one hour at room temperature. Sample was mounted with VECTASHIELD Vibrance Antifade Mounting Medium with DAPI (Vector Laboratories, Burlingame, CA, USA).

## 3. Results

This section may be divided by subheadings. It should provide a concise and precise description of the experimental results, their interpretation, as well as the experimental conclusions that can be drawn.

### 3.1. diffPCR

Figure 2 shows the HER2/IFNG ratios obtained upon diffPCR amplification of DNA samples of 30 healthy individuals. The cut-off point obtained, based on the mean value (0.73) and the standard deviation (0.25) is 1.22 (calculated as described above). All ratio values obtained above this threshold should be deemed positive. In this instance, only one healthy subject (number 15) yielded a ratio above the cut-off point.

Figure 2C reflects the results obtained with paired (tumoral and distal) tissue samples from the patients. DNA from eight tumoral samples (patients 3, 10, 11, 18, 19, 21, and 27) shows a HER2/IFNG ratio above the cut-off point aforementioned. The positive cell lines (BT-474 and SKBR-3) are well beyond this point, whereas the negative cell line (Caco-2) is below it. Finally, Figure 2E plots the overall results obtained with all samples: patients (tumoral and distal tissue), healthy control subjects and cell lines.

### 3.2. qPCR

Comparable results were obtained by the qPCR technique (Figure 2B). As before, the upper panel shows the HER2/IFNG ratios obtained upon amplification of DNA samples from the same 30 healthy individuals. With this technique, the mean value (0.68) and the standard deviation (0.39) obtained, yielded a cut-off point of 1.85. On this occasion, none of the controls studied yielded a value above the cut-off point.

When DNA from paired (tumoral and distal) samples from gastric adenocarcinoma patients are considered (Figure 2D), tumoral tissue samples from the same patients as above (3, 10, 11, 18 and 19), and none from distal tissue, yielded ratio results above this threshold. Control cell lines behave as before: BT-474 and SKBR-3 yielded positive results and Caco-2 a negative one. Figure 2F depicts results obtained with all samples used.

### 3.3. Immunohistochemistry and Immunofluorescence

Tissue staining, either by IHC or IF, confirmed the results previously obtained by diffPCR and qPCR in the samples analyzed (Figure 3). No HER2 staining was found in distal or tumoral, HER2 negative, samples from patients, whereas a clear staining was achieved in samples deemed positive by the techniques here explained.

As a control, a negative staining results along with a negative diffPCR and qPCR amplification was obtained, when a healthy or distal tissue from a patient was used (representative pictured, Healthy).

These results confirm that both PCR methodologies are suitable to detect HER2 amplification in patients with gastric adenocarcinoma.

## 4. Discussion

The determination of genetic amplifications of the HER2 gene continues to be of vital importance to determine patients with worse prognosis and susceptible to receive immunotherapy with the monoclonal antibody trastuzumab. In gastric cancer, the presence of HER2 gene amplifications is still a controversial issue, where there is no consensus on the frequency of patients presenting these amplifications, due to the difficulty in determining HER2 copy number variation by conventional methodologies (IHC and FISH).

We propose here a reliable methodology based on robust molecular genetics and a final scanning procedure to set up a cut-off point above which patients can be confidently considered HER2-positive. In particular, this methodology permits self-standardization for each laboratory, along with an easy assessment of in-house cut-off values by running an adequate number of control samples. The DNA-based approach is more stable and easier to interpret and has less interobserver variation. In addition, the procedure herein explained does not require any further standardization steps, and thus ease interlaboratory comparisons. It suffices for each laboratory to use DNA samples from healthy subjects to set-up their own cut-off point, and then run patients’ DNA. Positive or negative results are not subject to personal appreciation, nor they depend on the source of reagents (primary or secondary antibodies, fluorophores…) used in each laboratory.

With the methodologies herein proposed we detected HER2 amplification in 25% of patients, in keeping with published reports [22,23]. This result was corroborated by histological staining techniques (IHC and IF), the reliability of these techniques and the concordance of HER2 gene amplifications with the expression of this receptor in patients with gastric cancer. diffPCR methodology (including IFNG as control gene) has been successfully employed in previous works with other types of malignancies [19,24,25], although they lack the internal normalization we herein propose.

Of note, we report in this study different HER2 gene copy number variations in tumoral or distal tissue from the same patient with gastric cancer, both by diffPCR and qPCR (and confirmed by IHC and IF), suggestive of a tumoral tissue-specific event, rather than a pre-existing condition. We found no differences in the frequency of HER2 amplification between males and females, in keeping with previously published results using larger cohorts [26,27].

## 5. Conclusions

In summary, the approaches herein described may be reliable methods to assess HER2 copy number. A larger study, focusing on HER2 equivocal cases (assessed by IHC and FISH), may confirm this methodology as a new complementary assay to evaluate HER2 amplification in gastric cancer or other malignancies.

## Figures and Tables

**Figure 1 biology-10-00516-f001:**
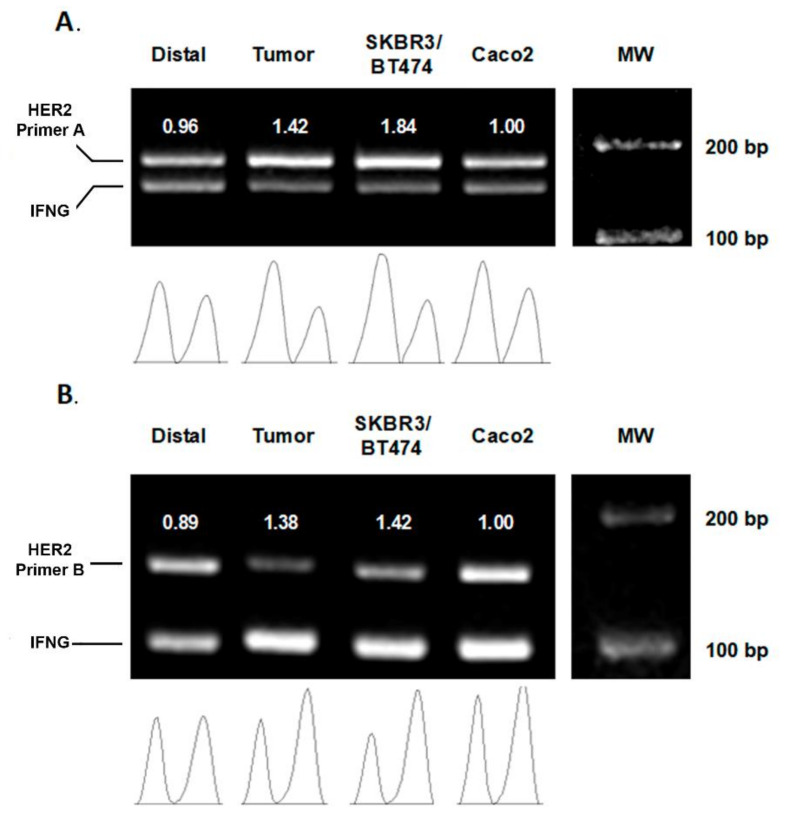
Representative agarose gel of diffPCR amplifications. (**A**) HER2 primers A (180 bp) and IFNG primers (150 bp), (**B**) IFNG primers (150 bp) and HER2 primers B (98 bp). Each PCR performed during the experiments included the amplification of paired samples (distal and tumor), a positive control (SKBR3 and BT474 DNA) and the negative control (Caco-2). Density graphs were quantified and relativized to Caco-2. The values below the bands correspond to the HER2/IFN-γ ratio. The results confirm that both PCR methodologies are suitable to detect HER2 amplification in patients with gastric adenocarcinoma. The full figure please see Appendix A.

**Figure 2 biology-10-00516-f002:**
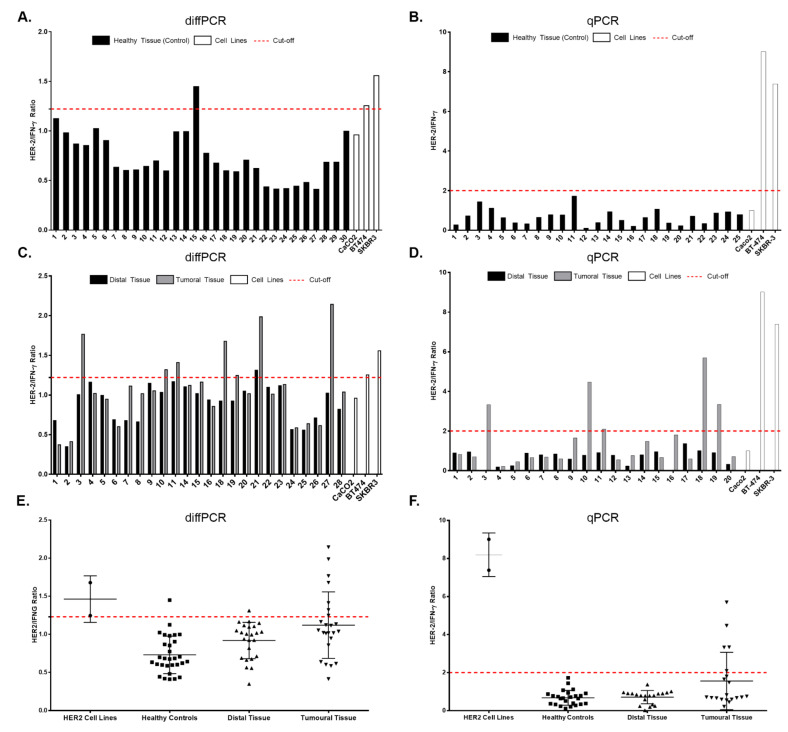
Quantification of HER2 amplification by diffPCR and qPCR. (**A**) diffPCR results obtained from 30 healthy control samples. All samples, except 15, are below the cut-off point (1.22) (**B**) qPCR results obtained from 30 healthy control samples. All samples are below the cut-off point (1.85). (**C**) diffPCR results obtained from paired samples (tumor and distal) from patients with gastric cancer. (**D**) qPCR results obtained from paired samples. Consistent results were obtained in patients 3, 10, 11, 18 and 19, being positive by both methodologies. (**E**) Graph show diffPCR results in paired samples from patients (tumoral and distal), healthy subjects and positive cell lines. (**F**) Graph show qPCR results in paired samples from patients (tumoral and distal), healthy subjects and positive cell lines.

**Figure 3 biology-10-00516-f003:**
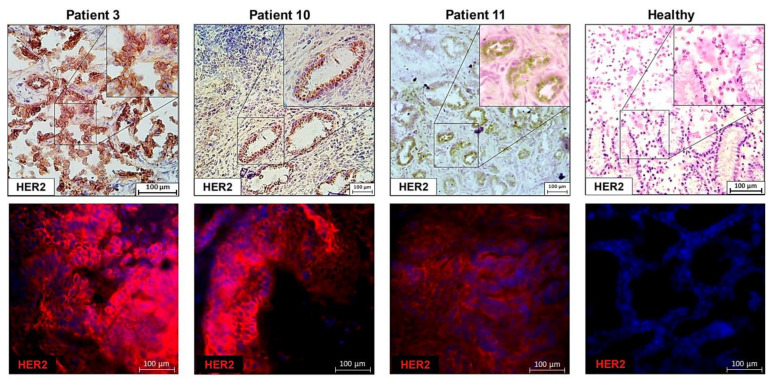
Immunohistochemistry (upper panels) and immunofluorescence (lower panels) showing positive HER2 staining in tumoral tissues from patients (3, 10 and 11) deemed HER2 positive by diffPCR and qPCR.

**Table 1 biology-10-00516-t001:** Patients’ demographic characteristics.

	Median	Range
Age	68	38–85
	n	(%)
**Sex**		
Male	16	(57.2)
Female	12	(42.9)
**UICC 7th edition TNM Staging**		
Stage I	9	(32.1)
Stage II	7	(25.0)
Stage III	4	(14.3)
Stage IV	8	(28.6)
**Treatment**		
Surgery	28	(100)
Chemotherapy *	28	(100)
Radiotherapy	28	(100)

*: Cisplatin/Oxaliplatin + 5-FU + anthracycline.

**Table 2 biology-10-00516-t002:** diffPCR primers and conditions. Italics represent one-per-cycle conditions.

Primers	Sequence (5′–3′)	Size (bp)	Cycles	Denaturation	Annealing	Elongation
HER2	AAGCATACGT	180	35	*94 °C*		70 °C
Primer A Fwd	GATGGCTGGT			*10* min ^1^	59.4 °C	1 min
HER2	CAATCTGCAT	180		94 °C	1 min	*70 °C*
Primer B Rvs	ACACCAGTTC			1 min		*10* min ^2^
HER2	CCTCTGACGT	98	32	*94 °C*		72 °C
Primer B Fwd	CCATCATCTC			*10* min ^1^	56 °C	1.5 min
HER2	ATCTTCTCGT	98		94 °C	1.5 min	*72 °C*
Primer B Rvs	GCCGTCGCTT			1.5 min		*10* min ^2^
IFNG	TCTTTTCTTTC	150	Shared by both diffPCR			
Primer Fwd	CCGATAGGT					
IFNG	CAGGGATGCT	150				
Primer Rvs	CTTCGACCTC					

^1^ Initial denaturation cycle, ^2^ Final elongation cycle (only 1 cycle per program).

## Data Availability

The data presented in this study are available on request from the corresponding author.

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
