# Peer review of "A Reliable and Standardizable Differential PCR and qPCR Methodology Assesses HER2 Gene Amplification in Gastric Cancer"

_biology, 2021, doi:10.3390/biology10060516_

Round 1

Reviewer 1 Report

The manuscript by Juarez et al describes two complementary PCR-based methods to determine HER2 amplification in gastric cancer. Identification of HER2 amplification is critical in gastric cancer patients since it determines further treatments, but it is difficult to assess which are the cases that are really amplified. In this study authors report two techniques that may be used to complement the current procedures, to better assess which are the amplified cases. The techniques are PCR-based and easy to implement. The results are clear and the manuscript is relevant and easy to follow. The discussion is not very extensive but puts the results well into context, however an explanation for the choice of IFNG to normalize HER2 levels needs to be provided.

Author Response

Thank you very much for your comments and for your interest in our results.

Regarding the use of IFNG as a control gene, this gene has been widely used as a control gene to assess HER2 amplification in breast cancer in previous works (Frye RA et al.  Oncogene 1989; Neubauer A et al. Oncogene 1992 and Valerón PF et al. Int JCancer 1996). The IFNG gene is located on chromosome 12, far from HER2 on chromosome 17, making it very unlikely they may share the same type of chromosomal alterations. Similarly, no major copy number alterations have been reported in the IFNG gene in gastric cancer. Although it is complicated to establish a perfect control due to the high number of genetic alterations in certain types of gastric cancer, we believe that IFNG is one of the most robust to achieve the proposed normalization.

In addition, and to refine the desired analysis, other control genes with no copy number variations described can be incorporated for this approach. in a clinical laboratory, which could refine the detection of these amplifications in complicated cases.

We have added a sentence explaining this point in the discussion section (lines 215 and 216 of the “revised manuscript - changes in red” document)

Reviewer 2 Report

Article entitled "A reliable and standardizable differential PCR and qPCR methodology assesses HER2 gene 2 amplification in gastric cancer," by Juarez et al., is a scientifically well presented article. 

However, I have following concerns:

  1. Abstract and background section needs to be more precise. Authors should provide literature and other evidence to support their hypothesis.
  2. It will be an excellent information to add in the very beginning of the article that why HER2 expression is so important in gastric cancer though only 25% of patients over-expressed HER2.
  3.  Expression pattern (similarities or differences if any or none) of HER2 in male and female subjects need to be elaborated.
  4. Figure legend of figure 3 is confusing. It is not clear when HER2- tissues were used when tissues from healthy controls were used. Please use one term consistently or make it clear. As patients may not express HER2 always. 
  5. If possible, use more clear pictures of ICC with scale bar in Figure 3.

Author Response

Thank you for your comments and suggestions.

  1. We added some references to show more evidence that support our hypothesis and work. They are now listed as references 4, 5, 6, 7 and 14.
  2. We added a sentence explaining the importance of HER2 genetic amplification and overexpression in gastric cancer (lines 39, 40, 45-49 of the “revised manuscript - changes in red” document), with the corresponding references (listed as 5, 6, 7 and 14.)
  3. In this work, we did not focus on the differences between male or females, as no difference in the HER2 genetic amplification between males and females in gastric cancer has been described (Marx, A.H. et al Human Pathology 2009; Ji, F. et al. World Journal of Gastroenterology 1999). We added a sentence explaining this point in the discussion section (lines 215 and 216 of the “revised manuscript - changes in red” document)
  4. Figure 3 legend has been updated to improve the understanding of the tissues used.
  5. IHC and IF pictures have been improved, and a 100 μm scale bar had been added on the right lower corner of the pictures.
